# Gastric Cancer Screening in Japan: A Narrative Review

**DOI:** 10.3390/jcm11154337

**Published:** 2022-07-26

**Authors:** Kazuo Yashima, Michiko Shabana, Hiroki Kurumi, Koichiro Kawaguchi, Hajime Isomoto

**Affiliations:** 1Division of Gastroenterology and Nephrology, Faculty of Medicine, Tottori University, 36-1 Nishicho, Yonago 683-8504, Japan; kurumi_1022_1107@yahoo.co.jp (H.K.); koichiro@tottori-u.ac.jp (K.K.); isomoto@tottori-u.ac.jp (H.I.); 2Sanin Rosai Hospital, 1-8-1 Kaikeshinden, Yonago 683-8605, Japan; shabana@saninh.johas.go.jp

**Keywords:** gastric cancer, gastric cancer screening, endoscopy, *H. pylori*, eradication therapy

## Abstract

Gastric cancer is the second leading cause of cancer incidence in Japan, although gastric cancer mortality has decreased over the past few decades. This decrease is attributed to a decline in the prevalence of *H. pylori* infection. Radiographic examination has long been performed as the only method of gastric screening with evidence of reduction in mortality in the past. The revised 2014 Japanese Guidelines for Gastric Cancer Screening approved gastric endoscopy for use in population-based screening, together with radiography. While endoscopic gastric cancer screening has begun, there are some problems associated with its implementation, including endoscopic capacity, equal access, and cost-effectiveness. As *H. pylori* infection and atrophic gastritis are well-known risk factors for gastric cancer, a different screening method might be considered, depending on its association with the individual’s background and gastric cancer risk. In this review, we summarize the current status and problems of gastric cancer screening in Japan. We also introduce and discuss the results of gastric cancer screening using *H. pylori* infection status in Hoki-cho, Tottori prefecture. Further, we review risk stratification as a system for improving gastric cancer screening in the future.

## 1. Introduction

Gastric cancer is the fifth most common cancer and the fourth leading cause of cancer-related deaths worldwide [1]. *Helicobacter pylori* (*H. pylori*) infection is considered the main cause of gastric cancer [2,3]. In Japan, the adjusted incidence and mortality rates of gastric cancer have decreased over the past few decades [4]. This decrease is mainly attributed to the reduction in *H. pylori* infection rates and the preventative effects of the *H. pylori* eradication therapy [5,6,7,8,9,10]. Despite this reduction, the number of gastric cancer cases ranks second and the number of deaths caused by gastric cancer ranks third in Japan [11], making it a critical public health problem.

In Japan, radiographic examination has been conducted since the 1960s as a secondary preventive measure for gastric cancer [12]. The revised 2014 Japanese Guidelines for Gastric Cancer Screening approved gastric endoscopy for use in population-based screening, together with radiography [13]. Currently, the government of Japan recommends either radiography or gastroscopic examination for gastric cancer screening [14]. However, there are some barriers, such as participation rate, endoscopic capacity, equal access, and cost-effectiveness [15,16,17,18].

Over 99% of gastric cancers in Japan are predisposed by a current or past *H. pylori* infection [19,20]. Furthermore, the background of gastric cancer risk has changed compared to the past due to the rapid decrease in the infection rate of *H. pylori* [5,6,7,8,9,10]. It has become necessary for efficient gastric cancer screening to classify patients as *H. pylori*-infected [8,21,22].

In recent years, image-enhanced endoscopy (IEE) [23], as well as artificial intelligence (AI), have been introduced in endoscopic diagnostics [24,25,26]. In this review, the present status and problems of gastric cancer screening in Japan are summarized. We present the results of gastric cancer screening using *H. pylori* infection status in Hoki-cho, Tottori prefecture. Further, we introduce risk stratification as a system for improving gastric cancer screening in the future.

## 2. Gastric Cancer in Japan

### 2.1. Epidemiology of Gastric Cancer

In Japan, gastric cancer accounted for almost half of all cancer deaths in the 1960s, but the proportion continues to decline. According to the 2021 cancer statistics forecast of the National Cancer Center Cancer-Information Service, “Cancer Registration and Statistics”, gastric cancer ranked third in the number of deaths after lung cancer and colorectal cancer. The total number of cancer deaths was 11.1% (42,000 people) [11]. The number of gastric cancer deaths has remained at 50,000 per year for the past few decades, and since 2011, it has been declining. However, more than 40,000 people lose their lives to stomach cancer every year. Gastric cancer has the second highest incidence rate at 12.9% (130,500 people), following colorectal cancer. As for the annual transition of gastric cancer, the age-standardized incidence and mortality are steadily decreasing, the number of cases is increasing, and the number of deaths tends to plateau due to an increase in the incidence and deaths caused by gastric cancer in the elderly population.

### 2.2. H. pylori and Gastric Cancer

The International Agency for Research on Cancer designated *H. pylori* as a clear gastric cancer carcinogenic factor (group 1) in 1994 [27] and recommended prevention by eradication in 2014 [28]. The presence of *H. pylori* infection is determined by histologic examination, the rapid urease test, serum antibody test, stool antigen test, or 13C-urea breath test. The effectiveness of the eradication treatment on gastric cancer prevention has been shown in a randomized controlled trial [29], and this primary preventive effect of the eradication of gastric cancer has been reported in recent meta-analyses [30,31,32]. Eradication of *H. pylori* reduces the risk of gastric cancer and mortality [33,34,35,36], but the risk still remains in the second decade after eradication [37]. Moreover, the pathogenicity and carcinogenicity of *H. pylori* depend on its strain. The East Asian type of *H. pylori*, which is popular in Japan, is more carcinogenic than the European-type *H. pylori* [38,39]. In addition, the presence of *H. pylori* with a positive babA2 gene may contribute to an increased risk of GC, especially in the Asian population [39,40]. In Japan, the eradication treatment for gastric and duodenal ulcers was covered by the National Health Insurance in 2000, and *H. pylori*-infected gastritis was added as an indication in 2013 [41]. According to recent reports in Japan, the risk of cumulative incidence of gastric cancer was 17.0% in men and 7.7% in women in the *H. pylori*-infected population, and <1% in the non-infected population [42]. More than 99% of gastric cancers in Japan are associated with *H. pylori*-infection gastritis [19,20]. Histopathological diagnosis of gastric cancer is performed according to the Japanese Classification of Gastric Carcinoma and the Vienna classification system [43,44]. Although gastric cancer that is not associated with *H. pylori* infection is extremely rare, gastric is cancer associated with autoimmune gastritis, gastric cancer due to *CDH1* gene mutation, fundic gland-type cancer, signet ring cell carcinoma, and cardia cancer are known [45]. Cardia cancer is often discovered at an advanced stage; thus, particular attention should be paid to it [46]. Moreover, the main risk factors of cardia cancer, which include gastroesophageal reflux disease and obesity, are different from those of gastric cancer associated with *H. pylori* [47].

As mentioned above, in Japan, the age-standardized incidence and mortality rate of gastric cancer has decreased over the past few decades due to a decrease in the incidence of *H. pylori* infection [4,5,6,7,8,9,10,11,12,13,14,15]. *H. pylori* infection rates in the 1960s, 1970s, and 1980s or later were 30%, 20%, and <10%, respectively [7]. A meta-analysis of the Japanese population shows that *H. pylori* infection rate is high in patients born in the 1940s; however, the infection rate decreased in patients who were born later, in the 1950s [9]. Although the morbidity rate of gastric cancer has continued to decrease due to the reduced *H. pylori* infection rates and the preventative effect of the *H. pylori* eradication therapy, the prevalence of *H. pylori* eradication has increased remarkably in recent years [8]. In the midst of dynamic changes in the incidence of *H. pylori* infection, it is considered to be important to pay attention to the high-risk groups in gastric cancer screening.

## 3. Gastric Cancer Screening Methods Used in Japan

### 3.1. Current Status and Problems of Upper Gastrointestinal Series

Annual radiographic screening for everyone >40 years of age in Japan was implemented in the 1960s as a secondary preventive measure for gastric cancer [12,14]. Gastric cancer screening using radiographic examination has proven to reduce mortality. It has an excellent mass-processing ability, and good accuracy, and is safe and cost-effective [48,49]. Furthermore, in recent case-control studies in Japan and South Korea, the effect of radiographic screening on mortality reduction was limited [50,51]. The Japan Society of Gastroenterological Cancer Screening formulated a revised version of the new gastric radiography guidelines (2011) [52]. The ability to view lesions by gastric radiographic examination has been greatly improved with the use of high-concentration, low-viscosity barium preparations and the advent of digital X-ray devices. Consequently, the rate of early detection of gastric cancer has exceeded 70% [53]. In addition, gastric cancer screening has been performed using imaging and AI to detect *H. pylori*-infected gastritis and gastric mucosal atrophy [54]. However, due to aging and immobilization of patients, radiation exposure, and lack of reading physicians and aging facilities, the rate of participation has been sluggish. Although endoscopic examinations have been approved by the revised 2014 Japanese Guidelines for Gastric Cancer Screening [13], it is impossible to replace all conventional radiography with endoscopic examinations due to problems relating to the capacity of endoscopy, budget, and access to examinees [14,15]. In population-based gastric cancer screening, it will be necessary to continue to utilize radiographic examinations with high processing capacity as a safety net.

### 3.2. Current Status and Problems of Upper Gastrointestinal Endoscopy

Radiographic examination is a screening method limited to Japan, but there is a growing international interest in endoscopic screening [55]. In Korea, in response to the results of domestic research, gastric cancer screening has been limited to endoscopic examinations [55,56].

In 2013, a case-control study was conducted in Japan and Korea. The research conducted in Japan involved a study on the population of Goto Islands in Nagasaki Prefecture [57] and a study on the population of Tottori Prefecture and Niigata City [58]. Although the sample size is small in the Nagasaki study, the mortality rate of gastric cancer was significantly decreased by 79% in participants of endoscopic screening (odds ratio [OR]: 0.206, 95% confidence interval [CI]: 0.044–0.965) [57]. In 2013, a case-control study that was conducted in Niigata City and four cities in Tottori Prefecture reported that the mortality rate was significantly lower by approximately 30% in people who underwent endoscopy 36 months before the date of gastric cancer diagnosis (OR: 0.695, 95% CI: 0.489–0.986) [58]. The studies that were conducted in Korea were large-scale research based on national databases. When the gastroscopic examination was performed even once in the past, the effect of reducing the gastric cancer mortality rate was confirmed to be 47% in individuals aged 40–74 years old (OR: 0.53, 95% CI: 0.51–0.56) [56]. Based on these results, a gastroscopy was recommended as a population-based screening method according to the revised 2014 Japanese Guidelines for Gastric Cancer Screening [13]. At the same time, it has changed from once a year for individuals aged >40 to once every 2 years for individuals aged >50 years, reflecting the recent decline in gastric cancer mortality by age group. In 2015, a study of Tottori Prefecture showed that endoscopic screening reduced the gastric cancer mortality rate by 67% compared with radiographic screening [50]. Zhang et al. conducted a meta-analysis that included 342,013 individuals in the six-cohorts and four-case-control studies that were previously published. This analysis demonstrated that endoscopic examination showed a 40% reduction in gastric cancer mortality rate (relative risk: 0.60, 95% CI: 0.49–0.73) [59].

According to reports from the area where endoscopic examinations were introduced, the gastric cancer detection rate was 0.05–0.32% for gastric X-ray examination and 0.30–0.87% for gastroscopic examinations [8,60]. Further, the gastric cancer detection rate of endoscopy was reported to have been approximately three times higher than that of X-ray examination. In Japanese studies, the proportion of early-stage cancer was approximately 70% in the radiographic screening group and >80% in the endoscopic screening group. Similarly, Hosokawa et al. previously reported that the detection rate of early cancer was higher in the endoscopic screening group than in the radiographic screening group [61]. However, the effectiveness of gastric cancer screening should be evaluated by the mortality reduction, and not by the detection rate.

Endoscopy can diagnose early-stage cancers that can be treated by endoscopic surgical dissection. Endoscopic surgical dissection has been performed for approximately half of early-stage cancers detected by endoscopic screening [62]. It seems to contribute to the maintenance of the quality of life after treatment. Moreover, recent development and widespread use of IEE and magnifying endoscopy have improved the endoscopic diagnosis of gastric cancer [23]. IEE is useful for diagnosing gastric cancer after eradication, which is usually difficult to detect [63]. In a recent study, we showed that photodynamic endoscopic diagnosis—based on the fluorescence of photosensitizers that accumulate in tumors—may be useful in the diagnosis of early gastric cancer regardless of the endoscopist’s experience and is useful for tumor detection; however, its usefulness has not been established because no prospective studies evaluating its usefulness have been performed [64].

As the participation rate in gastric cancer screening has decreased, its impact on mortality reduction has become limited. Although the participation rate in radiographic screening for gastric cancer has sunk below 10% [65], it is possible to improve the participation rate by introducing endoscopic screening as a method of gastric cancer screening. Notably, the participation rate is approximately 25% in municipalities that have already undergone endoscopic screening [66,67]. Thus, endoscopy is now the first choice for gastrointestinal tract examination instead of X-ray examination.

## 4. Risk Stratification for Gastric Cancer Screening

### 4.1. Risk Factors for Gastric Cancer

Risk factors for gastric cancer include *H. pylori* infection and accompanying gastric mucosal atrophy, smoking, and hereditary diseases, such as Lynch syndrome and familial adenomatous coli [23]. In addition, diet, lifestyle preferences, and Epstein-Barr virus infection have been reported as possible risk factors. Recently, it has been reported that approximately one-fifth of diffuse-type gastric cancers in Japan were attributable to the combination of alcohol intake and defective *ALDH2* allele or *CDH1* variants [68]. The most important method of obtaining information about these risk factors before endoscopic screening is a medical questionnaire. In addition, during the endoscopic examination, individuals can be stratified by gastric cancer risk based on *H. pylori* infection status and relevant findings suggestive of gastric cancer risk, as described in the endoscopy-based Kyoto classification of gastritis [69,70,71]. Endoscopic findings related to the risk of gastric cancer include moderate-to-severe gastric atrophy, enlarged gastric folds, nodular gastritis, xanthoma [72,73], and map-like redness [70]. As a result of examining the accuracy of *H. pylori* infection diagnosis by the “Kyoto classification of gastritis”, the sensitivity and specificity of detecting uninfected, existing infection, and current infection were 88.3% and 92.9%, 78.8% and 90.0%, and 67.1% and 91.4%, respectively. Moreover, risk classification by endoscopic examination was confirmed to have very high accuracy. However, to avoid false-negative results, an *H. pylori* antibody test was recommended [74].

### 4.2. Tests Used for Risk Stratification

According to the 2019 Basic Survey on National Life, 54.2% of men and 45.1% of women aged 40–69 years had undergone gastric cancer screening [75], approaching the target value of 50% of the 3rd Basic Plan for Cancer Countermeasures in Japan. However, in recent years, the number of *H. pylori*-negative people has increased, and the gastric cancer-adjusted mortality rate has naturally decreased [5,6,7,8,9,10,11]; following this, there has been a problem with cost-effectiveness in the strategy of simply increasing the participation rate. In the future, it may be necessary to stratify individuals according to gastric cancer risk by determining risk factors—such as a history of *H. pylori* infection and gastric mucosal atrophy—and reflect them in the selection of endoscopy and the determination of the screening interval.

The “ABC method”, a combined assay for serum anti-*H. pylori* IgG antibody and serum pepsinogen (PG) levels, is generally used in Japan as a gastric cancer risk classification system [76]. Itoh et al. reported a strong correlation between the ABC classification system and radiological findings in relation to the risk of gastric cancer [77]. However, the revised 2014 Japanese Guidelines for Gastric Cancer Screening do not recommend this method due to insufficient scientific evidence regarding its effectiveness in gastric cancer screening [13]. The risk of gastric cancer can be stratified based on factors, such as the presence of *H. pylori* infection and the extent and severity of gastric atrophy. The serum anti-*H. pylori* IgG antibody titer can predict an individual’s *H. pylori* infection status, whereas its titers vary greatly depending on the test kit used.Serum PG levels reflect the status of gastric mucosal inflammation and serve as a marker for atrophic gastritis. Individuals with PG I levels of ≤70 ng/ml and PG I/II ratio of <3 are classified as PG test positive, and people with a history of *H. pylori* eradication, treatment of proton pump inhibitors, previous gastric resection and impairment of renal function are excluded to ensure correct stratification. This method classifies individuals into the following four groups according to their serological status: (1) group A, anti-*H. pylori* IgG antibody (−)PG (−); (2) group B, anti-*H. pylori* IgG antibody (+)/PG (−); (3) group C, anti-*H. pylori* IgG antibody (+)/PG (+); and (4) group D, anti-*H. pylori* IgG antibody (−)/PG (+), which also included those with autoimmune gastritis (type A gastritis) [76]. Notably, a meta-analysis conducted by Terasawa et al. demonstrated that groups A, B, and C + D were significantly different in their respective gastric cancer risk [78]; thus, this stratification is expected to serve as a mass screening system for this disease.

As the development of gastric cancer in patients not infected with *H. pylori* is extremely rare in Japan, it may be expected that the *H. pylori*-uninfected population could be excluded from the mass screening system for gastric cancer. However, group A included patients with a high risk of developing gastric cancer and could not be regarded as truly *H. pylori*-negative [79,80]. The presence of *H. pylori*-infected individuals in group A is a crucial problem because the individuals are wrongly considered to have an extremely low risk for gastric cancer, similar to healthy, *H. pylori*-uninfected individuals. The endoscopic grade of atrophy is an accurate predictive marker for gastric cancer [81,82]. To exclude individuals who are truly *H. pylori*-negative, an endoscopic evaluation of the gastric mucosa should be performed [83,84]. It is inefficient to perform endoscopy in all patients as this is expensive and requires high manpower of endoscopists.

According to a report by the Kanazawa City Medical Association [84], gastric cancer may develop at an annual rate of 0.31% in a state with advanced atrophy (O-3) classified by Kimura and Takemoto [85], and it is possible to stratify the risk of gastric cancer using endoscopic diagnosis. Therefore, endoscopic diagnosis of atrophy may be more effective than the ABC classification system for predicting the risk of gastric cancer.

Several cost-effectiveness analyses demonstrated that endoscopic surveillance is a cost-effective method to reduce gastric cancer mortality. A comprehensive systematic review showed that endoscopic screening is cost-effective in high-incidence countries, and that targeted endoscopic screening of high-risk populations is also generally cost-effective in low-intermediate incidence countries [86]. Recently, Kowada et al. demonstrated that biennial endoscopy for patients with mild-to-moderate gastric mucosal atrophy and annual endoscopy for patients with severe gastric mucosal atrophy were the most cost-effective measures after *H. pylori* eradication [87].

### 4.3. Gastric Cancer Screening Tests Performed at Hoki-cho, Tottori Prefecture

Since 2000, patients in Tottori Prefecture were able to select between endoscopic and radiographic examinations. The rate of gastric cancer screening by endoscopic or radiographic examination in Hoki-cho, Tottori Prefecture has remained around 20%, which is not sufficient, as the national target is 50%. With the aim of accelerating endoscopic screening and eradication therapy for *H. pylori* infection, Hoki-cho in Tottori Prefecture has implemented a risk evaluation system for gastric cancer for 5 years since 2014 by testing the serum for *H. pylori* antibodies [88]. Target populations included individuals aged 20 and 35–70 years in each year, and who underwent at least one examination through the evaluation system during this period (Figure 1).

In cases with negative results for *H. pylori* diagnosis, we incorporated the serum PG method. During the 5 years from 2014 to 2018, there were a total of 6191 target individuals, of whom 2464 were screened (participation rate: 39.8%). The total number of *H. pylori*-positive cases was 753 (30.6%), and that of cases negative for *H. pylori* antibody and positive for the PG method was 58 (2.4%). The frequency of *H. pylori* positivity was 9.2% in individuals aged 20 years and <40% in individuals aged 60–70 years. This gradually increased with advancing age (Figure 2). The rate was highest (38.4%) among patients aged 60–70 years of age.

Consequently, during the 5-year study period, 71.3% of the examinees underwent a detailed endoscopic examination (Table 1), and two patients with early gastric cancer were detected. Eradication therapy was implemented in 97.6% of cases that had a positive result for *H. pylori* infection after undergoing a detailed endoscopic examination. On the other hand, only 33.7% and 22.8% of individuals with positive screening results in 2014 and 2015, respectively, had received a periodic endoscopic screening at least once during the three years after the following year. Therefore, it is important to increase the participation rate of this project and the rate of detailed endoscopic examinations to further increase in the detection of the risk of gastric cancer and implement periodic endoscopic screening.

The rate of population-based gastric cancer screening in Hoki-cho was 20.6% in 2013; however, after the introduction of the *H. pylori* infection screening, it increased to 26.2% in 2015, 22.8% in 2016, 23.2% in 2017, and 24.3% in 2018. In 2018, 657 (63.4%) of the 1036 patients had opted for endoscopic examination (26.1% in 2013, 35.3% in 2014, 52.9% in 2015, 50.7% in 2016, and 57.0% in 2017), contributing to the steady increase in the use of endoscopy (Table 2).

This implies that screening using the *H. pylori* antibody test is useful for improving the rate of participation and efficient gastric cancer endoscopy. In the future, it will be necessary to verify the effect of reducing gastric cancer mortality by combining *H. pylori* antibody testing and endoscopic examination and to implement the optimal screening interval for each *H. pylori*-infected and uninfected person. In addition, it is important to improve the true rate of participation by recommending endoscopic examination to those who require it.

## 5. Future Directions for Gastric Cancer Screening

### 5.1. Optimal Age and Intervals for Screening

According to Japan’s national screening program, the recommended age for gastric cancer screening was changed to >50 years due to a decrease in the incidence of gastric cancer in 40-year-olds [13]. Similarly, the British Society of Gastroenterology guidelines suggested endoscopy screening be considered in individuals aged >50 years with multiple risk factors for gastric adenocarcinoma (male, smokers, and pernicious anemia) [89]. In Korea, gastric cancer screening is conducted for populations aged 40–74 years [55]. A study in Japan based on nationwide data showed that the endoscopic screening program would be cost-effective when implemented for populations aged 50–75 years [90]. A nationwide study in Singapore revealed that gastric cancer screening was cost-effective when used among Chinese men aged 50–70 years [91].

A different screening interval might be defined and considered depending on its relationship to the individual’s background and gastric cancer risk. The incidence of gastric cancer differs according to individual risks and is mainly defined by *H*. *pylori* infection status and atrophic gastritis. In Korea, an interval of 2 years is recommended [92]. The British Society of Gastroenterology recommends that endoscopic follow-up should be performed every 3 years for individuals with severe chronic atrophic gastritis or intestinal metaplasia, and within one-year intervals for low-grade intraepithelial neoplasia—similar to the management of epithelial precancerous conditions and lesions in the stomach (MAPS II) guideline [93]. In Japan, high-grade intraepithelial neoplasia should be treated clinically. The national program in Japan recommends repeated gastric cancer screening every 2–3 years [14]. However, high-quality prospective research is required to determine the optimal follow-up interval for endoscopic screening in Japan. If individuals with a low risk of gastric cancer could be identified and adopted in the screening programs, their screening interval could be expanded. Hamashima et al. introduced infection atrophy diagnosis using endoscopy and serological testing or risk stratification and conducted a nationwide prospective study to set the interval between risk-specific screenings [17]. It is expected that the results of this research will reduce the burden on patients by appropriately classifying the risk of gastric cancer and extending the interval between screenings for low-risk patients. The research also aims to establish a system that enables the target population to access endoscopic screening fairly by effectively utilizing limited medical resources.

### 5.2. AI as a New Screening Method

In gastric cancer screening, both radiographic and endoscopic examinations may be eluded by gastric cancer [56,94,95]. In population-based screening, the specialist is required to carry out a double check, the labor is intensive, and the evaluation of the accuracy is difficult. Recently, diagnosis of *H. pylori* infection and detection of gastric cancer using AI have been reported. The sensitivity and specificity of endoscopic *H. pylori* infection diagnosis were 81.9% and 83.4% using AI, 79.0% and 83.2% by an average endoscopist, and 85.2% and 89.3% by an endoscopic specialist, respectively [96]. On the other hand, when AI detection was conducted in three groups. That is, *H. pylori*-positive, *H. pylori*-negative, and eradicated *H. pylori*, the rate of correct diagnosis decreased to 77% [97]; hence, there is room for further improvement in diagnosis using AI, including that of cases following *H. pylori* eradication. AI has a high sensitivity for gastric cancer, but its positive predictive value is low [24,25,26]. However, this has rapidly improved [98]. In addition to its accuracy, AI diagnostic imaging is expected to reduce the burden of double-checking and effectively extract patients who need follow-up endoscopy [98]. It is expected that intervention of gastric cancer screening using AI may reduce gastric cancer deaths more efficiently than the conventional methods of screening.

## 6. Conclusions

While endoscopic gastric cancer screening has been initiated nationwide in Japan, the incidence of *H. pylori* infection has decreased and the number of cases following *H. pylori* eradication has increased. Moreover, the importance of ABC classification reflecting *H. pylori* infection status and gastric atrophy before endoscopic screening is being increasingly recognized. Considering its cost-effectiveness, spreading the use of endoscopic screening is desirable to establish a new medical examination provision system that conducts examinations at appropriate screening intervals, according to the individual’s background and risks.

## Figures and Tables

**Figure 1 jcm-11-04337-f001:**
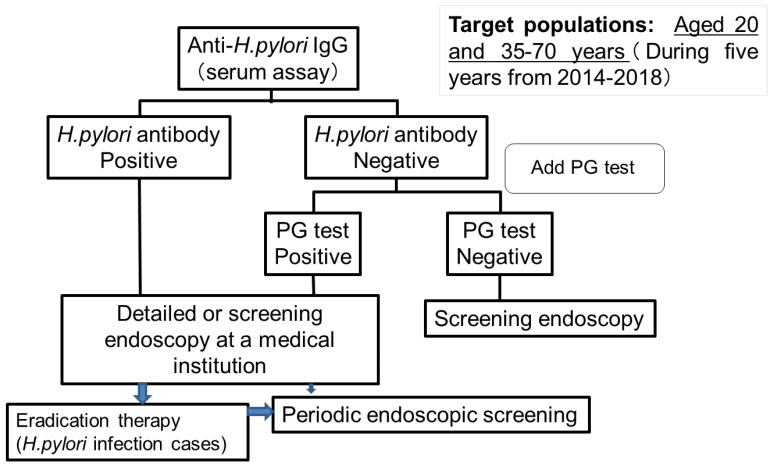
Flow chart of *H. pylori* antibody test project in Hoki-cho, Tottori prefecture. Individuals with PG I levels of ≤70 ng/mL and PG I/II ratio of <3 are classified as PG test positive, which is equal to gastric atrophy. PG, pepsinogen.

**Figure 2 jcm-11-04337-f002:**
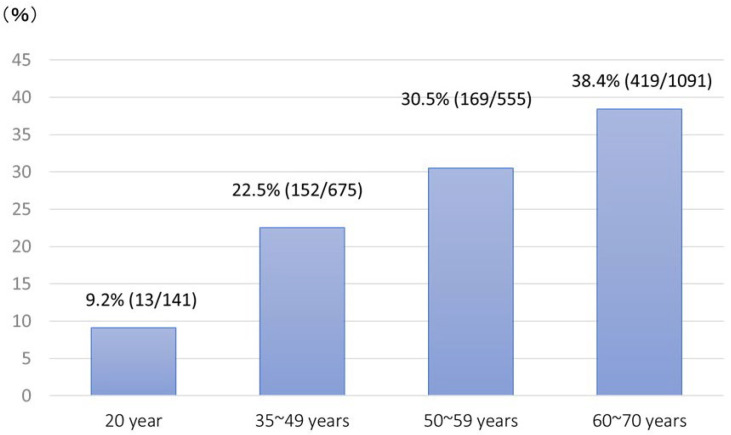
The frequency of *H. pylori* positivity according to age (2014~2018).

**Table 1 jcm-11-04337-t001:** Results of *H. pylori* antibody test project in Hoki-cho, Tottori Prefecture.

Year	2014	2015	2016	2017, 2018	Total
Examinees (*n*)	910	776	311	467	2464
Cases requiring detailed endoscopy (*n*)	323	259	109	121	811
Examination required rate (%)	35.4	33.4	35.0	25.9	32.9
Cases undergone screeningendoscopy (*n*)	258	181	61	78	578
Examination rate (%)	79.9	69.9	56.0	64.5	71.3

**Table 2 jcm-11-04337-t002:** Annual trends in the rate of participation for population-based gastric cancer screening in Hoki-cho, Tottori Prefecture.

Year	2013	2014	2015	2016	2017	2018	2019
Target population (*n*)	4533	4533	4533	4257	4257	4257	4257
Examinees (*n*)	934	963	1188	970	986	1036	1039
Participation rate (%)	20.6	21.2	26.2	22.8	23.2	24.3	24.4
Proportion of endoscopy among gastric cancer screening tests (%)	26.1	35.3	52.9	50.7	57.0	63.4	65.9

The data were obtained from “Cancer Screening Report in Tottori Prefecture”.

## Data Availability

Data will be available from the corresponding author upon reasonable request.

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
