# Peer review of "Gastric Cancer Screening in Japan: A Narrative Review"

_jcm, 2022, doi:10.3390/jcm11154337_

Round 1
Reviewer 1 Report
In the present review article, the authors discussed gastric cancer screening in Japan. They also introduced their own experience in Hoki-cho, Tottori prefecture, and indicated some potential avenues for future investigations. Interestingly, they included a section about artificial intelligence (AI). The idea is of high interest in the field of prevention and diagnosis, the selection of cited studies looks unbiased, and the manuscript is well-written, especially in terms of flow and consistency.
Comments:
Line 56: I think it is better to put a dash (-) in the middle of “National Cancer Center Cancer Information Service” and change it to “National Cancer Center-Cancer Information Service” if the name remains correct regarding its Japanese term.
Line 70-72: “… and this primary preventive effect of the eradication of gastric cancer has been reported in a recent meta-analysis [30-32]”. Three references are provided although the statement is talking about one meta-analysis, as is understood from “a” in “a meta-analysis”.
Line 100, 2.3. Current status and problems of gastric radiographic examination: The statement “Gastric cancer screening using radiographic examination has proven to reduce mortality. It has an excellent mass-processing ability, good accuracy, and is safe and cost-effective [45,46]” are based on old references. However, at the end of this section, the authors state that “Furthermore, in recent case-control studies in Japan and South Korea, the effect of radiographic screening on mortality reduction was limited [50,51]” which is based on more recent references. I suggest moving the latter statement (based on newer papers) to the end of the former statements.
Line 127, a study in “Goto Islands in Nagasaki Prefecture [54]” is mentioned but this study remained undiscussed throughout the section.
Line 143: “… meta-analysis that included 342,013 individuals in the six-cohort and four-case-control studies …”. I think it should be “… six cohorts and four case-control …”.
Line 175: 2.4. Current status and problems of gastroscopic examination: at the end of the section, “It is worth noting that attention should be paid to severe adverse events” remains undiscussed. I wonder what adverse events the authors are talking about.
Author Response
Thank you for your comments on our manuscript entitled " Gastric cancer screening in Japan: a narrative review
The comments are very helpful for revising and improving our paper, as well as the important guiding significance to other research.
Replies to the reviewers’ comments:
Reviewer 1:
Comments:
Line 56: I think it is better to put a dash (-) in the middle of “National Cancer Center Cancer Information Service” and change it to “National Cancer Center-Cancer Information Service” if the name remains correct regarding its Japanese term.
Response:
Thank you for checking our mistake. We changed to “the National Cancer Center Cancer-Information Service”.
Line 70-72: “… and this primary preventive effect of the eradication of gastric cancer has been reported in a recent meta-analysis [30-32]”. Three references are provided although the statement is talking about one meta-analysis, as is understood from “a” in “a meta-analysis”.
Response:
Thank you for checking our mistake. We changed “a recent meta-analysis” to “recent meta-analyses”.
Line 100, 2.3. Current status and problems of gastric radiographic examination: The statement “Gastric cancer screening using radiographic examination has proven to reduce mortality. It has an excellent mass-processing ability, good accuracy, and is safe and cost-effective [45,46]” are based on old references. However, at the end of this section, the authors state that “Furthermore, in recent case-control studies in Japan and South Korea, the effect of radiographic screening on mortality reduction was limited [50,51]” which is based on more recent references. I suggest moving the latter statement (based on newer papers) to the end of the former statements.
Response:
In the section of “2.3. Current status and problems of gastric radiographic examination”, we moved the statement “Furthermore, in recent case-control studies in Japan and South Korea, the effect of radiographic screening on mortality reduction was limited.”, which is based on more recent references, to the end of the statements “Gastric cancer screening using radiographic examination has proven to reduce mortality. It has an excellent mass-processing ability, good accuracy, and is safe and cost-effective [45,46]”.
Line 127, a study in “Goto Islands in Nagasaki Prefecture [54]” is mentioned but this study remained undiscussed throughout the section.
Response:
We added the following sentence to the section of “2.4 Current status and problems of gastroscopic examination”
“Although the sample size is small in the Nagasaki study, the mortality rate of gastric cancer was significantly decreased by 79% in participants of endoscopic screening (odds ratio [OR]: 0.206, 95% confidence interval [CI]: 0.044-0.965).”
Line 143: “… meta-analysis that included 342,013 individuals in the six-cohort and four-case-control studies …”. I think it should be “… six cohorts and four case-control …”.
Response:
Thank you for checking our mistake. We changed “six cohorts”.
Line 175: 2.4. Current status and problems of gastroscopic examination at the end of the section, “It is worth noting that attention should be paid to severe adverse events” remains undiscussed. I wonder what adverse events the authors are talking about.
Response:
We thought this sentence was unnecessary and deleted it.
Reviewer 2 Report
The manuscript by Yashima et al. summarizes the current status and problems of gastric cancer screening in Japan and the highlights the needing of risk stratification as a system for improving gastric cancer screening in the future. The Authors also introduced and discussed
the results of gastric cancer screening using H. pylori infection status in Hoki-cho, Tottori prefecture.
Although the work is well organized and comprehensively described, it is not scientifically sound.
Moreover, the description of histologic detection of H.P. is missing as well as the Japan criteria to make a histological diagnosis of gastric cancer which are differen from the Europe and America ones. In this respect a comparison with Western data is required.
Author Response
Thank you for your comments on our manuscript entitled " Gastric cancer screening in Japan: a narrative review
The comments are very helpful for revising and improving our paper, as well as the important guiding significance to other research.
Replies to the reviewers’ comments:
Reviewer 2:
Although the work is well organized and comprehensively described, it is not scientifically sound. Moreover, the description of histologic detection of H.P. is missing as well as the Japan criteria to make a histological diagnosis of gastric cancer which are differen from the Europe and America ones. In this respect a comparison with Western data is required.
Response:
We added the following sentence and reference in the section “2.2 H. pylori and gastric cancer in Japan”.
“The presence of H. pylori infection is determined by histologic examination, the rapid urease test, serum antibody test, stool antigen test, or 13C-urea breath test.”
“Histopathological diagnosis of gastric cancer is performed according to the Japanese Classification of Gastric Carcinoma and the Vienna classification system.”
- Japanese Gastric Cancer Association. Japanese classification of gastric carcinoma-3rd English edition. Gastric Cancer 2011, 14, 101-112.
- Schlemper R.J.; Riddell R.H.; Kato Y.; Borchard F.; Cooper H.S.; Dawsey S.M.; Dixon M.F.; Fenoglio-Preiser C.M.; Fléjou J.F.; Geboes K.; et al. The Vienna classification of gastrointestinal epithelial neoplasia. Gut 2000, 47, 251–255.
Author Response
Thank you for your comments on our manuscript entitled " Gastric cancer screening in Japan: a narrative review
The comments are very helpful for revising and improving our paper, as well as the important guiding significance to other research.
Replies to the reviewers’ comments:
Reviewer 3:
Comments: Because of complexity of the analyzed aspects in some parts of the article there is a little mess and the reader can be a little bit lost. Sometimes there is a need to read fragments at least twice.
Response:
We correct complexity of in the section of “2.3 Current status and problems of gastric radiographic examination”.
Specific comments:
Line 23 - H. pylori should be written in italics -
Response:
Thank you for checking our mistake. We changed to “H. pylori”.
Line 56 - the National Cancer Center Cancer Information Service - is it really ok with this fragment? If not - should be clarified -
Response:
Thank you for checking our mistake. We changed to “the National Cancer Center Cancer-Information Service”.
Lines 73-74 -the authors mention that pathogenicity and carcinogenicity of H. pylori depend on its strain, that the East Asian type of H. pylori is more carcinogenic than the European type. It’s true. However, maybe it is worth to add at this point that Asian population is more predisposed for being infected because of possessing specific receptors (more than other populations) in their stomachs (importance of mucins!), e.g. Lewis b antigens which are proved receptors for bacterial adhesins.
Response:
We added the following sentence and reference in the section of “2.2. H. pylori and gastric cancer in Japan”.
“In addition, the presence of H. pylori with positive babA2 gene may contribute to increased risk of GC, especially in Asian population.”
- Association of Helicobacter pylori babA2 gene and gastric cancer risk: a meta-analysis. Kpoghomou M.A.; Wang J.; Wang T.; Jin G.BMC Cancer 2020, 20, 465. doi: 10.1186/s12885-020-06962-7
Lines 103-104 -The authors write: “Gastric cancer screening using radiographic examination …. is safe …”. However, in lanes 111-112 they write: “However, due to …..radiation exposure …, the rate of participation has been sluggish”. In my opinion, to some degree, there is a kind of incompatibility - should be clarified
Response:
In the section of “2.3. Current status and problems of gastric radiographic examination”, we moved the statement “Furthermore, in recent case-control studies in Japan and South Korea, the effect of radiographic screening on mortality reduction was limited.”, which is based on more recent references, to the end of the statements “Gastric cancer screening using radiographic examination has proven to reduce mortality. It has an excellent mass-processing ability, good accuracy, and is safe and cost-effective [45,46]”.